# Enhanced Energy-Storage Performances in Sodium Bismuth Titanate-Based Relaxation Ferroelectric Ceramics with Optimized Polarization by Tuning Sintering Temperature

**DOI:** 10.3390/ma15144981

**Published:** 2022-07-18

**Authors:** Jianhua Wu, Ziyue Ma, Yuan Yao, Ningning Sun, Ye Zhao, Yong Li, Runchang Liu, Xihong Hao

**Affiliations:** 1Inner Mongolia Key Laboratory of Ferroelectric-Related New Energy Materials and Devices, School of Materials and Metallurgy, Inner Mongolia University of Science and Technology, Baotou 014010, China; wjh2219333820@126.com (J.W.); ya746108363@163.com (Z.M.); yaoyuan@binn.cas.cn (Y.Y.); sunning@imust.edu.cn (N.S.); yzhao@imust.edu.cn (Y.Z.); 2Yunnan Longivy Technology Co., Ltd., Kunming 650217, China

**Keywords:** Na_0.5_Bi_0.5_TiO_3_, sintering temperature, energy-storage properties, dielectric ceramics

## Abstract

Energy-storage capacitors based on relaxation ferroelectric ceramics have attracted a lot of interest in pulse power devices. How to improve the energy density by designing the structure of ceramics through simple approaches is still a challenge. Herein, enhanced energy-storage performances are achieved in relaxation ferroelectric 0.9 (0.94Na_0.5_Bi_0.5_TiO_3_-0.06BaTiO_3_)-0.1NaNbO_3_ (NBT-BT-NN) ceramics by tuning sintering temperature. The original observation based on Kelvin probe force microscopy (KPFM) presented that the sintering temperature has a key effect on the electrical homogeneousness of the ceramics. It is found that a high electrical homogeneousness can induce quick and active domain switching due to the weakening of the constraint from built-in fields, resulting in a big polarization difference. This work provides a feasible strategy to design high-performance energy-storage ceramic capacitors.

## 1. Introduction

In recent years, with the energy structure gradually transforming, ceramic capacitors have received considerable attention due to their high-power density, good thermal stability, and fast charge–discharge speed [1,2,3]. However, the low energy-storage density has become a major obstacle for its practical applications [4,5]. Therefore, improving the energy-storage density and efficiency of ceramic capacitors is a current research hotspot [6,7]. Generally, the total energy storage (*W*), the recoverable energy-storage density (*W_rec_*), the loss energy density (*W_loss_*), and energy efficiency (*η*) of dielectric capacitors can be calculated and analyzed by the following formulas [8,9]:(1)$$W_{\;}=\;\int_{0}^{P_{max}}EdP$$
(2)$$W_{rec\;}=\int_{P_{r}}^{P_{max}}EdP$$
*W_loss_* = *W* − *W_rec_*
(3)
(4)$$\eta =\frac{W_{rec}}{W}\;\times \;100\%=\frac{W_{rec}}{W_{rec\;}{+W}_{loss}\;}\;\times \;100\%$$
where *E*, *P_max_* and *P_r_* express the applied electric field, maximum polarization and remnant polarization, respectively. As shown in the schematic diagram in Figure 1, excellent energy-storage properties can be achieved by improving *P_max_*, reducing *P_r_* and increasing the electric breakdown strength (*E_b_*) at the same time [10,11,12].

In previous studies, researchers used chemical modifications and structural design to improve energy-storage performance and have achieved some good results. For instance, Yang et al. found that grain boundary diffusion could be prevented by doping elements; thus, effectively inhibiting grain growth. On this basis, by introducing BiFeO_3_ into K_0.5_Na_0.5_NbO_3_ and reducing the grain size from 5.4 to 0.15 µm, 0.9K_0.5_Na_0.5_NbO_3_-0.1BiFeO_3_ ceramics with *W_rec_* of 2 J cm^−3^ at 206 kV cm^−1^ were successfully obtained [13]. Compared with macroscopic ferroelectric domains, PNRs have lower switching barriers and higher dynamic activity under external electric fields. Therefore, the energy-storage ceramics constructed by PNRs unfold higher *W_rec_*, slimmer hysteresis loops, and higher efficiency, which attract the attention of researchers. An effective strategy to achieve ultrahigh energy-storage performance via nano-scale polarization mismatch was proposed by Hu’s team. They fabricated 0.6BaTiO_3_-0.4Bi(Mg_1/2_Ti_1/2_)O_3_ ceramics and obtained a high *W_rec_* of 4.49 J cm^−3^ and *η* of 93% [14]. The researchers also took advantage of the strong anisotropic of the perovskite structure by adjusting its orientation to achieve better performance. Bai and his partners fabricated <001>-oriented 0.94BNT–0.06BT ceramics using the template grain growth method, and realized *W_rec_* of 1.6 J cm^−3^ with excellent frequency stability (0.1~100 Hz) and temperature stability (25~120 °C) [15]. Synthetic of two-phase composite materials is also an effective strategy to improve energy-storage density. For example, core-shell structures are widely used in the design of ferroelectric composites. Xu et al. fabricated the BaTiO_3_@SiO_2_ ceramics which obtained 4.799 J cm^−3^ recoverable energy storage at 370 kV cm^−1^, in which the SiO_2_ coating prevents the growth of the grain and makes the structure compact; thus, reducing the leakage current to obtain good energy-storage performance [16].

According to the above research, it is feasible to improve the energy-storage performance of materials by composition doping or structural design [17,18]. However, these methods have many uncertain factors and large instability, so there is still a long way to go to be put into practice [19,20,21]. Therefore, a simple, stable, and easy to implement method needs to be explored. In this work, 0.9 (0.94Na_0.5_Bi_0.5_TiO_3_-0.06BaTiO_3_)-0.1NaNbO_3_ (NBT-BT-NN) ferroelectric ceramics were prepared by a traditional solid-state sintering method. Exploring the influence of different sintering temperatures on the dielectric properties, and energy-storage behavior for proving that optimizing the sintering temperature, is a simple and stable method to improve the energy-storage performance of materials.

## 2. Materials and Methods

In this work, a series of NBT-BT-NN ferroelectric ceramics (sintering at 1000, 1050, 1100, 1150 and 1200 °C, abbreviated as *C*_1000_, *C*_1050_, *C*_1100_, *C*_1150_ and *C*_1200_) were prepared by a traditional solid-state sintering method.

According to formula 0.9 (0.94Na_0.5_Bi_0.5_TiO_3_-0.06BaTiO_3_)-0.1NaNbO_3_, various oxides and carbonate (Sinopharm Chemical Reagent Co., Ltd., Shanghai, China) of TiO_2_ (98.0%), Bi_2_O_3_ (99.2%), Nb_2_O_5_ (99.9%), Na_2_CO_3_ (99.8%), and BaTiO_3_ (99.0%) were weighed and mixed for the synthesis of ceramic powders.

The mixed raw materials were added with anhydrous alcohol for 24 h of ball milling, and the resulting slurries were dried and ground into powders. Then, it was heated to 800 °C in a muffle furnace and kept for 2 h. After grinding to powder, anhydrous alcohol was added and mixed again in a ball mill for 24 h and dried. Subsequently, the pellets were prepared in an electric powder pressing machine with PVA solution as the binder. Ceramics are obtained by sintering pellets in a muffle furnace at the above temperature for 2 h. Finally, the polished ceramic piece was plated with a gold electrode with a 2 mm diameter for dielectric and ferroelectric properties testing.

The phase composition of the sintered NBT-BT-NN ceramics was characterized by X-ray diffraction (XRD, Bruker D8 Advanced Diffractometer, German). The diffractometer used a Cu target, the X-ray scanning speed is 4 °/min, and the scanning angle is selected as 20°~70°. The surface morphology of the ceramics was conducted by the scanning electron microscopy (FE-SEM, ZEISS Supra 55, German). Polarization-electric field loops of the ceramics were characterized by a ferroelectric analyzer (Radiant Technologies, Inc., Albuquerque, NM). The domains of ceramic were observed by the piezoelectric force microscope (PFM, Bruker, Icon). The microscopic electrical properties of ceramic surfaces were tested using a surface Kelvin probe force microscopy (KPFM, Bruker, Icon). The dielectric property was measured by a computer-controlled LCR meter (TH2828, Tongue, China).

## 3. Results and Discussion

The XRD patterns of NBT-BT-NN ceramics were displayed in Figure 2. A main perovskite phase was formed for all the ceramics, and a small quantity of the pyrochlore phase was observed. The sharp diffraction peak and the relatively large diffraction intensity indicate a high crystallinity, and NN has good solid solubility in NBT-BT ceramics [22]. Therefore, the NBT-BT-NN ceramics can be synthesized in a wide temperature range with high quality, which will be beneficial to the testing of their dielectric and ferroelectric properties.

Figure 3a–e illustrate SEM images of NBT-BT-NN ceramics. It can be seen from the figure that with the increase in sintering temperature, the density of ceramics increases first, and then decreases. Among them, *C*_1050_ is the densest. The grain distribution was presented in the inset of Figure 3a–e. Compared with the other ceramics, the *C*_1050_ exhibits a small grain size and a concentrated distribution, indicating outstanding uniformity. For further comparison, the average grain size of the *C*_1050_ is 0.79 µm, which is the minimum among all ceramics. Small grain size is beneficial to the energy-storage performances of ceramic materials [23].

The frequency dependence of dielectric constant (*ε_r_*) of NBT-BT-NN ceramics from 10 kHz to 1 MHz at room temperature was displayed in Figure 4a. The *ε_r_* of all sintering temperatures decrease with increasing frequency. This is mainly because the change of its internal electric dipole is affected by the test frequency under the electric field, and the *ε_r_* is affected accordingly [24]. With the increase in frequency, the switching speed of the electric dipole cannot keep up with the turning speed of the applied electric field gradually, which makes the intensity of polarization weaken correspondingly. These changes also indicate that the dielectric constant is highly dependent on sintering temperature [25]. In the meantime, with the sintering temperature rising from 1000 °C to 1200 °C, the *ε_r_* increases firstly, and then decreases, and the maximum value is obtained at the sintering temperature of 1050 °C. Figure 4c shows the *ε_r_* comparison at 10 kHz. The *ε_r_* of the ceramics is 1599, 2208, 2072, 1763 and 1698 at different sintering temperatures from 1000 to 1200 °C at 10 kHz, respectively. The results prove that the sintering temperature of 1050 °C can enhance dielectric properties of the ceramics.

Figure 4b shows the temperature dependence of *ε_r_* of NBT-BT-NN ceramics with different sintering temperatures at 1 MHz. Typical characteristics of relaxor behavior, namely broadened dielectric peaks, can be observed over the measured temperature range of 50 to 250 °C [26,27]. The temperature *T_m_*, corresponding to the maximum *ε_r_*, is related to the thermal relaxation of PNRs and the transition from rhombohedral (*R3c*) PNRs to tetragonal (*P4bm*) PNRs [28]. The variation trend of *T_m_* for each ceramic is shown in Figure 4d: *T_m_* moves toward the lower temperature range as the sintering temperature increases. This phenomenon indicates an increased degree of diffusion phase change, and ceramics show stronger relaxation behavior. Generally speaking, dielectric materials with relatively large *ε_r_* often show excellent energy-storage behavior, so the NBT-BT-NN *C*_1050_ are expected to obtain a relatively large energy-storage density.

To investigate the energy-storage performance of NBT-BT-NN ceramics with different sintering temperatures, the variations of *P-E*, *W_rec_* and *η* under different electric fields at room temperature and 100 Hz are shown in Figure 5. With the increase in the electric field, the polarization increases, and the *P-E* loops gradually show a saturation state. It can be calculated that the energy-storage density of all NBT-BT-NN ferroelectric ceramics increases as the electric field increases. However, the electric field will also increase the *W_loss_*, which leads to the reduction of *η*. Among them, when the electric field increases from 20 to 120 kV cm^−1^, the *W_rec_* of the ceramic with a sintering temperature of 1050 °C obtains the largest increase compared with the other ceramics, increasing from 0.04 to 1.42 J cm^−3^, and maintains a high *η* of 71%. This further proves that 1050 °C is a suitable sintering temperature for NBT-BT-NN energy-storage ceramics.

Figure 6a displays the *P-E* loops of the NBT-BT-NN ceramics at different sintering temperatures and Figure 6b shows their corresponding values of *P_max_−P_r_*. The *P-E* loops were measured at room temperature under the field of 120 kV cm^−1^ at the frequency of 100 Hz. It can be clearly seen that all ceramics show typical hysteresis loops of relaxor ferroelectric, the polarization is close to saturation under high electric field, and all maintain slimmer shapes. Among them, the ceramics with a sintering temperature of 1050 °C have the largest *P_max_*. The results of *P_max_−P_r_* have an important impact on the energy-storage performance of dielectric energy-storage ceramics, and its values are calculated and indicated in Figure 6b. The *P_max_−P_r_* values of ceramics sintered at 1000~1200 °C were 27.5, 30.3, 29.6, 29.4, and 28.2 µC cm^−2^, respectively. It can be seen that the *C*_1050_ has the maximum *P_max_−P_r_* value, which is beneficial to obtain a larger *W_rec_*. Figure 6c shows the *W_rec_* and *η* of ceramics at different sintering temperatures. The *W_rec_* of ceramics sintered at 1000~1200 °C are 1.35, 1.42, 1.38, 1.37 and 1.32 J cm^−3^, respectively. The corresponding *η* are 73%, 71%, 70%, 69% and 68%. As expected, the sintered ceramics with the maximum *P_max_−P_r_* value at 1050 °C showed the highest energy-storage density of 1.42 J cm^−3^, and achieved a large energy-storage efficiency. That is to say, the sintering temperature of ceramics has a great impact on *W_rec_* and *η*. According to the comparative results, sintering at 1050 °C is the optimal sintering schedule.

Changes in dielectric behavior and energy-storage performance are usually closely related to domain morphology and its dynamic response to the external electric field. PFM is an effective tool to deeply study the micromorphology of materials [29]. The domain structure and evolution process of *C*_1050_ and *C*_1200_ after poling with −200 V electric field are shown in Figure 7a–f. Figure 7a,d show the initial states of the two ceramics when polarized: strip regions with alternating light and dark can be seen in both ceramics, which belong to the long-range ordered ferroelectric domain morphology. As the voltage is removed, the domains are gradually switched [30]. The *C*_1050_ had only a small number of domains that did not switch after 15 min of rest, and finally recovered to the initial state completely after 30 min. Correspondingly, as shown in Figure 7e, most of the domains of *C*_1200_ were still in the polarized state after 15 min, and finally returned to the unpolarized state after 30 min. Compared with other ceramics, the internal electric field of *C*_1050_ has less hindrance to domain switching, and the response speed of domains to the external electric field is fast. Most of the domains can quickly return to an unpolarized state. This corresponds to the slimmer *P-E* loops and smaller *P_r_* of the ceramics, thus achieving better energy-storage performance.

Figure 8a–e show the surface potential and the corresponding potential distribution statistics of NBT-BT-NN ceramics with different sintering temperatures obtained by KPFM. The different colors in the figures indicate the potential distribution on the ceramic surface. Figure 8a,e show a large range of their potential distribution, with multiple peaks appearing. Figure 8b has only one peak, which is relatively sharp. This indicates that the *C*_1050_ has a more uniform potential. The wide potential distribution may cause the formation of multiple energy barriers in the ceramics, and then cause the generation of internal electric fields [31]. These internal electric fields inhibit the switching of the domains to some extent, resulting in the decrease of *P_max_* and the increase of *P_r_*, and producing a smaller *P_max_−P_r_* value. The uniform potential indicates the high electrical uniformity, which has little effect on the domain switching under the external electric field. Therefore, the *C*_1050_ have a larger *P_max_−P_r_* value.

Figure 9a shows the temperature dependence of the *P-E* hysteresis loops of *C*_1050_ from 20 °C to 180 °C at 60 kV cm^−1^. It can be seen that the ceramics exhibit typical ferroelectric hysteresis loops at all test temperatures, and the *P_r_* and the coercive field (*E_c_*) of the ceramics decrease with the increasing temperature. The decrease of *P_r_* is caused by the decrease in overall polarization strength of dielectric material as a result of the increase in temperature. At the same time, with the increase in test temperature, the interface energy of the ferroelectric domain decreases gradually, which easily leads to the change of domain wall movement, and thus leads to the reduction of *E_c_* [32]. Moreover, the *P-E* loops gradually become slimmer and slimmer with the increments of test temperature, and reach the thinnest state at 100 °C, which is conducive to obtaining a larger energy-storage density and efficiency. Figure 9b shows the variation of energy-storage density and efficiency with the test temperature corresponding to Figure 9a. At 20 °C, the *W_rec_* of NBT-BT-NN ferroelectric ceramic is 0.4 J cm^−3^, and the corresponding energy-storage efficiency is 66%. With the increase of the test temperature, the energy-storage density and efficiency showed a trend of first increasing and then decreasing, and the maximum energy-storage density was obtained when the test temperature was 100 °C. At 180 °C, the *W_rec_* is 0.43 J cm^−3^, and the energy-storage efficiency reaches 78%. Compared with the test result at room temperature, the energy-storage density and efficiency increased by 7.5% and 18%, respectively. The above results show that NBT-BT-NN ferroelectric ceramics have better temperature stability of energy-storage behavior in the temperature range of 20~180 °C.

## 4. Conclusions

In conclusion, 0.9 (0.94Na_0.5_Bi_0.5_TiO_3_-0.06BaTiO_3_)-0.1NaNbO_3_ (NBT-BT-NN) ceramics were prepared by the traditional solid-state method, and the effects of different sintering temperatures on dielectric properties and energy-storage performance were investigated. All ferroelectric ceramics are highly crystalline and exhibit predominantly perovskite phases. The *C*_1050_ has the best dielectric properties, the ferroelectric properties are stable under different test electric fields and a large *W_rec_* of 1.42 J cm^−3^ together with the moderate *η* of 71% at 120 kV cm^−1^ was obtained. Based on the analysis of KPFM test data, the *C*_1050_ achieved outstanding performance due to its rapid switching of electric domain and uniform surface potential. In addition, the ceramic exhibits excellent temperature stability in a wide temperature range of 20~180 °C. Therefore, NBT-BT-NN ferroelectric ceramics sintered at 1050 °C are promising candidates for pulsed power capacitors applications. Moreover, it is proven that the method to obtain excellent energy-storage performance by adjusting sintering temperature is stable and easy to realize.

## Figures and Tables

**Figure 1 materials-15-04981-f001:**
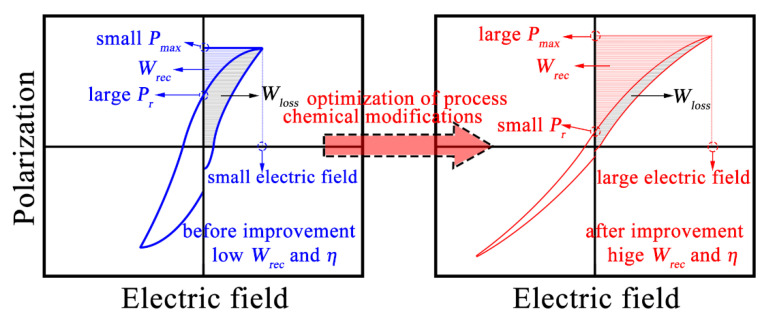
Schematic diagram of *P-E* loops for realizing superior energy-storage properties.

**Figure 2 materials-15-04981-f002:**
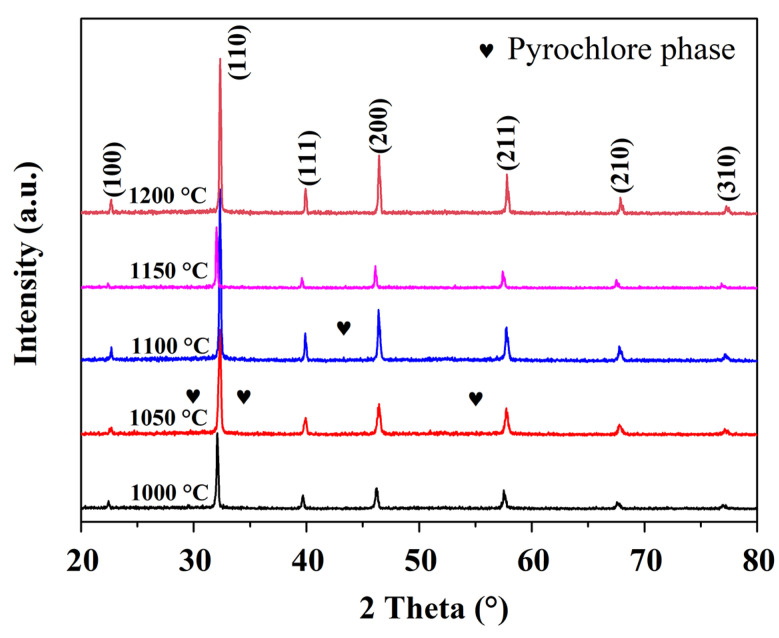
X-ray diffraction patterns of NBT-BT-NN ceramics in a wide range from 20° to 70°.

**Figure 3 materials-15-04981-f003:**
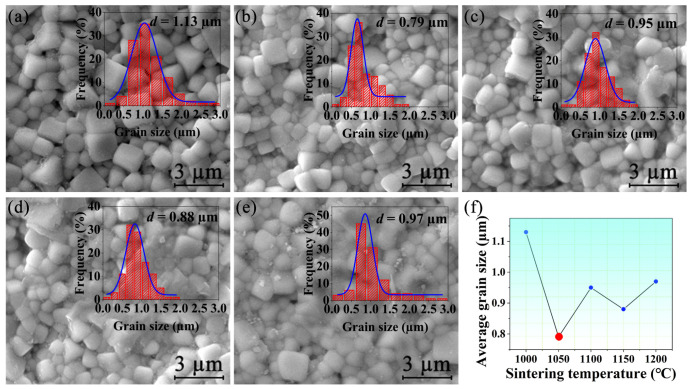
SEM images of the NBT-BT-NN ceramics with sintering temperature of (**a**) 1000 °C, (**b**) 1050 °C, (**c**) 1100 °C, (**d**) 1150 °C, (**e**) 1200 °C. The insets in (**a**–**e**) are the corresponding grain size distribution. (**f**) Variation of average grain size of ceramics at different sintering temperatures.

**Figure 4 materials-15-04981-f004:**
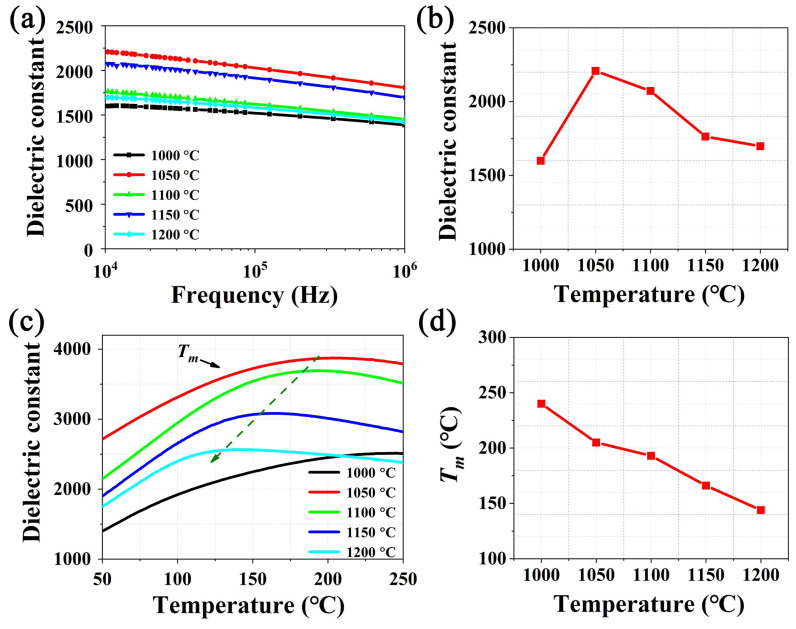
(**a**) Frequency dependent of the dielectric constant and (**b**) the corresponding change at 10 kHz for NBT-BT-NN ceramics. (**c**) Temperature dependence of dielectric constant and (**d**) the variation trend of *T_m_* for each ceramic.

**Figure 5 materials-15-04981-f005:**
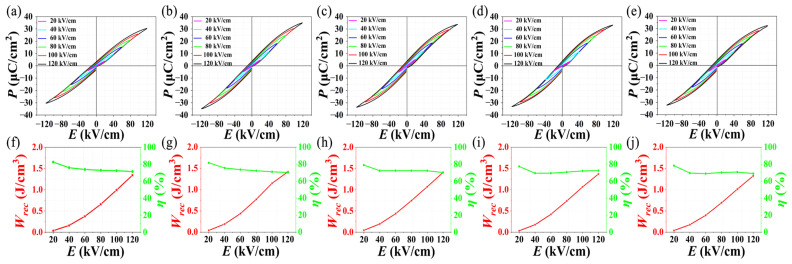
Room temperature *P-E* loops under different sintering temperatures for the NBT-BT-NN ceramics (**a**) 1000 °C, (**b**) 1050 °C, (**c**) 1100 °C, (**d**) 1150 °C, (**e**) 1200 °C, (**f**–**j**) shows the variation of *W_rec_* and *η* as a function of the applied field.

**Figure 6 materials-15-04981-f006:**
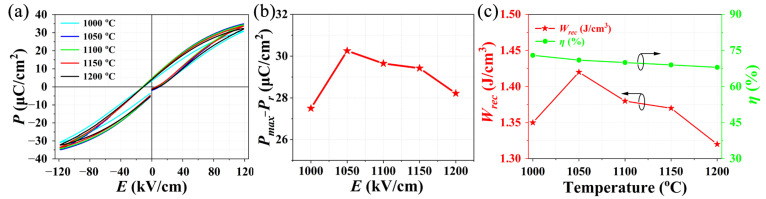
(**a**) Room temperature *P-E* loops of NBT-BT-NN ferroelectric ceramics at different sintering temperatures. (**b**) *P_max_−P_r_* values of each ceramic. (**c**) The *W_rec_* and *η* of NBT-BT-NN ferroelectric ceramics at different sintering temperatures.

**Figure 7 materials-15-04981-f007:**
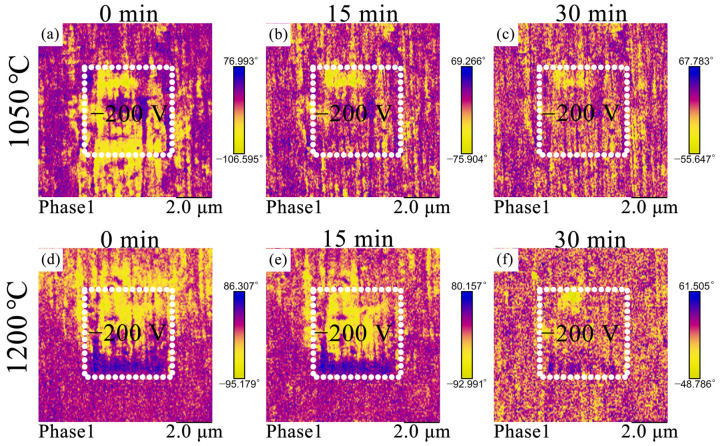
PFM image of NBT-BT-NN ceramics at different sintering temperatures (**a**–**c**) 1050 °C, (**d**–**f**) 1200 °C.

**Figure 8 materials-15-04981-f008:**
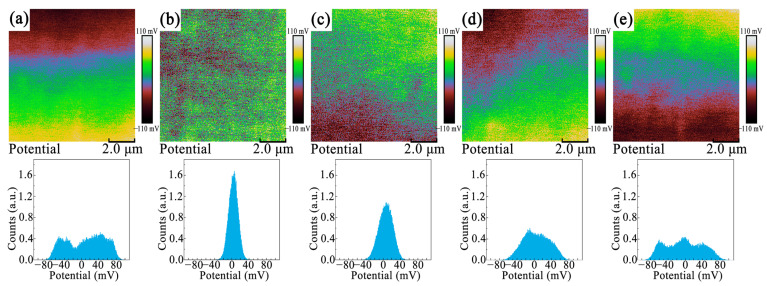
The surface potential maps and corresponding potential profile of NBT-BT-NN ceramics at different sintering temperatures (**a**) 1000 °C, (**b**) 1050 °C, (**c**) 1100 °C, (**d**) 1150 °C, (**e**) 1200 °C.

**Figure 9 materials-15-04981-f009:**
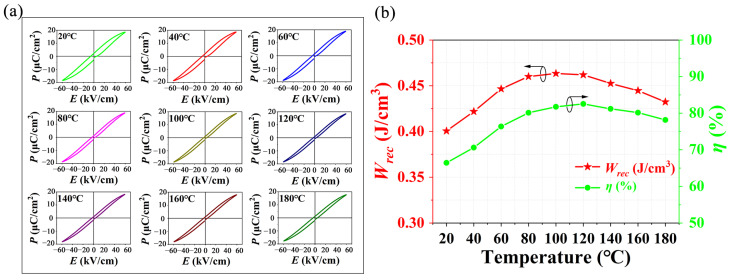
(**a**) *P-E* loops of NBT-BT-NN ceramic which were measured at different test temperatures. (**b**) Temperature dependence of *W_rec_* and *η* of NBT-BT-NN ceramic.

## Data Availability

The data presented in this study are available on request from the corresponding author.

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
