# Peer review of "Enhanced Energy-Storage Performances in Sodium Bismuth Titanate-Based Relaxation Ferroelectric Ceramics with Optimized Polarization by Tuning Sintering Temperature"

_materials, 2022, doi:10.3390/ma15144981_

Round 1

Reviewer 1 Report

The paper presents interesting research and the results of the experiment. The article requires significant corrections and additions (there are many bugs), and in my opinion, it can be published after major revision and redrafting the paper.

Detailed comments are below.

line 26 “[6, 7].  Generally,” unnecessary double space. 

lines 26, 27, 28, 29, 30, 31, 32, 34, 35, 40, 46, 47, 50, 107, 108, 110, 116, 120, 123, 124, 125, 134, 136, 138, 139, 141, 144, 146, 147, 148, 152, 154, 156, 157, 159, 160, 162, 165, 166, 181, 193, 194, 196, 201, 203, 204, 207, 208, 211, 215, 221, 222, 229 - all symbols should be written in italics. 

line 32 is “Where” should be “where”.

lines 46, 191 is “ceramic” should be “ceramics”.

line 65/66 missing free line.

line 67 is “Sintering” should be “sintering”.

line 67 “…at 1000, 1050, 1100, 67 1150 and 1200” - missing “°C”. 

line 67 is “1200. Abbreviated as…” should be “1200°C, abbreviated as…”.

line 68 the authors did not provide the sintering time of ceramic samples.

line 68 “C1200.”  unnecessary dot. 

line 67 is “1200. Abbreviated as…” should be “1200°C, abbreviated as…”.

lines 104, 105, 145, 184, 185, 200 -  the “°C” has a larger font.

line 147 “BaTiO3(99.0%)” - missing space. 

lines 89/90 “All ceramics show 89 single perovskite phase, and no secondary phase was observed”. The waveforms presented in the Fig.1 have too thick a line, which makes it difficult to detect additional peaks, which are still visible for individual waveforms. e.g.: 1000°C - about 29.5°, 31°, 1050°C - about 22°, 24°, 27°, 29.5°, 31°, 51°, 56°, 1100°C - about 44°, 1150°C - about 54°. They may indicate the presence of a foreign phase. Especially since some of them appear in the area of the angles for the pyrochlore phase. This should be checked and corrected.

Fig.1 Low resolution of the figure, too thick waveform lines, unnecessary space before “°C”. 

lines 97/98 “All the ceramics exhibit equably arranged grains with clear boundaries, and no obvious pores are observed.” The statement is not true because, the SEM photos posted contradict it, i.e.: “…with clear boundaries” - the statement does not apply to the samples from Fig.(d) and (e), “…no obvious pores” - the statement does not apply to the sample from Fig.(a). Moreover which means the sentence: “All the ceramics exhibit equably arranged grains…” if, on the SEM microstructures we see a large heterogeneity in the grain size?

line 103 Figure 2f shows that the largest grains occur for the C1000 sample i.e. sample with the lowest sintering temperature. It is commonly known that an increase in temperature increases the grain growth. How do the authors explain the reduction of the ceramic grain with the increase of the sintering temperature?

Fig.2 “d” should be written in italic. 

lines 118/121 The comparisons of the permittivity results were made for the high frequencies of 10 kHz and 1 MHz. Why were such high frequencies chosen, and not e.g. the standard frequency 1 kHz? After all, as the frequency increases, the permittivity decreases and the phase transition becomes even more blurred.

line 122 Why was the research started from 50°C and ended at 250°C? Why were measurements not made from room temperature to higher temperatures?

lines 123, 125 “Tc” - is the Curie phase transition temperature. With this type of diffused phase transition fuzzy, the temperature Tm for em (maximum of permittivity) should be used. 

Fig.3 Low resolution of the figure, the graphs inside the figures are too small and completely unreadable. Besides, it is difficult to find the temperature of the phase transition here. The measuring range should be wider and the graphs should be compiled for a lower frequency (e.g. 1 kHz).

line 131 is “…constant of NBT-BT-NN ceramics. (b) Temperature…” should be: “…constant and  (b) temperature…”

line 135 Why was the research made the tested at 100 Hz? Have the authors tried to take measurements at lower frequencies? 

line 138 The authors mention about “dielectric loss”, but why is the temperature measurement of this parameter not shown? This should be completed. 

Fig.4 Low resolution of the figure, the charts inside the figures are too small and completely unreadable and all symbols should be written in italics. Moreover, the result for 120 kV/cm is presented only in Fig.(b), in the others it is not.

How thick were the ceramic samples that even 120 kV/cm could be applied? Are the values presented in the P-E graphs calculated taking into account the thickness? The P-E graphs show that probably not.

Fig.5 Low resolution of the figure, the graphs inside the figures are too small and all symbols should be written in italics.  

Fig.7 Low resolution of the figure, the diagrams below are too small and completely unreadable.  

Fig.8 Low resolution of the figure, the graphs inside the figures are very, very tiny and all symbols should be written in italics. The sample for which the temperature loops were made should be specified.

lines 226-228 “All ferroelectric 226 ceramics exhibit highly crystalline and a single perovskite structure without second phase generation.” - comment as above.

References: 24% of the literature items are self-citations [1], [4], [5], [19], [23], [25], [28].

Reviewer 2 Report

The paper: Enhanced energy-storage performances in NBT-based relaxation ferroelectric ceramics with optimized polarization by tuning sintering temperature refers to enhanced energy-storage performances achieved in relaxation ferroelectric 0.9(0.94Na0.5Bi0.5TiO3-0.06BaTiO3)-0.1NaNbO3 (NBT-BT-NN) ceramics by tuning sintering temperature. These ceramics were prepared by the traditional solid-state method, and the effects of different sintering temperatures on dielectric properties and energy storage performance were investigated. The materials sintered at 1050°Care promising candidates for pulsed power capacitors applications. Some minor corrections are necessary. I do not understand the next sentence used in the abstract: “The original observation based on Kelvin probe force microscopy (KPFM) presented that the sintering temperature has a key effect on electrical homogeneousness of the ceramics.” You did not use this method for the characterization of your study. It must be removed or replaced.

Reviewer 3 Report

This paper reports on the Enhanced energy-storage performances in NBT-based relaxation ferroelectric ceramics with optimized polarization by tuning sintering temperature”. Introduction and conclusion, methodology and reference, results and discussion seems be corrected.

I have few comments to the manuscript:

1.     All manuscript. Corrected from e.g. “[4,5]” to “[4-5]”.

2.     The acronym “BaTiO3@20wt%SiO2” it is correct?

3.     Materials and methods. Content too compact, separate the materials used, preparation, and research methods in each paragraph.

Taking into account all comments the manuscript may be published in Materials after minor revision.

Reviewer 4 Report

First time a term appears in the text it must be explained even it is very (for example, NBT-BT-NN, PEM). Also, avoid abbreviation in the title. In equations (1)-(3) is not indicate the Wloss calculation formula. The Introduction was a rich contain of recent bibliography. At the beginning of the Section 2, we have mention sintering the sample of different values, but the quantity is not indicated – I suppose is the temperature.  In general, figures might be enlarged to be clearer. For example, picture-in-picture situation from Figures 3 and 4 are hard to see. It is not clear in Figure 4 and 5 how the Wrec and Wloss are identify from the measurements. Please extend this section. The Conclusion section must be enlarged.

Round 2

Reviewer 1 Report

I recommend the article for publication

Reviewer 4 Report

Thank you for considering all my remarks. I have no other comments.